# Transcriptional phenotype of the anti-parasitic benzodiazepine meclonazepam on the blood fluke *Schistosoma mansoni*

Clair R. Henthorn[1☯], Paul McCusker[2,3☯], Winka Le Clec'h[4], Frédéric D. Chevalier[4], Timothy J.C. Anderson[5], Mostafa Zamanian[1], John D. Chan[1,2,6]*

1 Department of Pathobiological Sciences, University of Wisconsin-Madison, Madison, Wisconsin, United States of America, 2 Department of Cell Biology, Neurobiology & Anatomy, Medical College of Wisconsin, Milwaukee, Wisconsin, United States of America, 3 Institute for Global Food Security, School of Biological Sciences, Queen's University Belfast, Belfast, United Kingdom, 4 Disease Intervention & Prevention Program, Texas Biomedical Research Institute, San Antonio, Texas, United States of America, 5 Disease Intervention and Prevention program, Texas Biomedical Research Institute, San Antonio, Texas, United States of America, 6 Global Health Institute, University of Wisconsin-Madison, Madison, Wisconsin, United States of America

☯ These authors contributed equally to this work.
* jchan32@wisc.edu

## Abstract

There are limited control measures for the disease schistosomiasis, despite the fact that infection with parasitic blood flukes affects hundreds of millions of people worldwide. The current treatment, praziquantel, has been in use since the 1980's and there is a concern that drug resistance may emerge with continued monotherapy. Given the need for additional antischistosomal drugs, we have re-visited an old lead, meclonazepam. In comparison to praziquantel, there has been relatively little work on its antiparasitic mechanism. Recent findings indicate that praziquantel and meclonazepam act through distinct receptors, making benzodiazepines a promising chemical series for further exploration. Previous work has profiled the transcriptional changes evoked by praziquantel treatment. Here, we examine in detail schistosome phenotypes evoked by *in vitro* and *in vivo* meclonazepam treatment. These data confirm that meclonazepam causes extensive tegument damage and directly kills parasites, as measured by pro-apoptotic caspase activation. *In vivo* meclonazepam exposure results in differential expression of many genes that are divergent in parasitic flatworms, as well as several gene products implicated in blood feeding and regulation of hemostasis in other parasites. Many of these transcripts are also differentially expressed with praziquantel exposure, which may reflect a common schistosome response to the two drugs. However, despite these similarities in drug response, praziquantel-resistant parasites retain susceptibility to meclonazepam's schistocidal effects. These data provide new insight into the mechanism of antischistosomal benzodiazepines, resolving similarities and differences with the current frontline therapy, praziquantel.

**Data availability statement:** All relevant data are within the manuscript and its Supporting Information files. FASTQ files containing RNA-Seq data are deposited in the NCBI SRA database under bioproject accession numbers PRJNA1150591 and PRJNA1177852 with the biosample accession numbers: SAMN43291307, SAMN43291308, SAMN43291309, SAMN43291310, SAMN43291311, SAMN43291312, SAMN43291313, SAMN43291314, SAMN43291315, and SAMN43291316.

**Funding:** This work was supported by funding from NIH-NIAID R21AI146540 (JDC & MZ), R01AI151171 (MZ), R01AI123434 and R01AI133749 (TJCA), the UW-System Regent Scholar Award (JDC). Fellowship funding for CRH was provided through the NIH Parasitology and Vector Biology Training grant (T32AI007414). The funders had no role in study design, data collection and analysis, decision to publish, or preparation of the manuscript.

**Competing interests:** The authors have declared that no competing interests exist.

## Author summary

Schistosomiasis, a neglected tropical disease caused by parasitic flatworms, affects hundreds of millions of people worldwide. The drug praziquantel is the frontline treatment for this disease, but alternative therapies are needed to address the potential emergence of drug resistance. In this study, we revisit an old antischistosomal compound, meclonazepam, which belongs to a chemical series (benzodiazepines) which have promise for further development as schistosomiasis therapies. Following meclonazepam treatment we were able to observe biochemical readouts of worm death (pro-apoptotic caspase activation), indicating effective schistocidal activity. Drug-evoked changes in gene expression were then measured, showing that many transcripts affected by meclonazepam exposure are unique to parasitic flatworms. Many gene products with differential expression also have putative direct or indirect interactions with the host immune system. Finally, we tested meclonazepam's ability to kill both praziquantel-susceptible and praziquantel-resistant populations of schistosomes. The drug was equally effective at killing both groups of parasites, suggesting that benzodiazepines could be developed as valuable alternative therapies in the fight against schistosomiasis.

## Introduction

Schistosomiasis is currently controlled with just one anthelmintic drug, praziquantel (PZQ), that has been in use for over four decades [1]. Reliance on a single therapy is concerning given the potential for emerging drug resistance [2,3]. One class of compounds that have received renewed interest in recent years as antischistosomal leads are benzodiazepines. The antischistosomal activity of benzodiazepines was discovered by Roche in the late 1970's [4], approximately the same time as Bayer and Merck developed PZQ. Anthelmintics in use prior to that period had significant drawbacks, such as efficacy against limited parasite species and potential for toxicity (e.g., organophosphates or antimony compounds). Studies on rodent models of schistosome infection demonstrated that the benzodiazepine meclonazepam (MCLZ, also referred to as 3-methylclonazepam and Ro 11–3128) was efficacious against the African parasite species that account for the vast majority of human infections (*Schistosoma mansoni* and *Schistosoma haematobium*) [4]. A small scale human trial confirmed this efficacy [5], but unfortunately also revealed dose-limiting sedative side effects [6]. Meanwhile, PZQ proved broad-spectrum, safe, and effective, and has become the frontline therapy for treating numerous flatworm parasite infections.

While PZQ is an effective antischistosomal treatment with high egg reduction rates (95%) and cure rates (57% to 88%) [7], there are several reports of PZQ treatment failure in humans [8,9] and against veterinary flatworm parasites [10,11]. PZQ-resistant parasites have been reproducibly selected numerous times in laboratory settings [12–14], and there is standing genetic variation at the putative PZQ receptor which indicates that mass drug administration programs using PZQ monotherapy could select for resistance in the field [3]. Therefore, we are interested in re-visiting MCLZ and other benzodiazepines to better understand their antiparasitic mechanism of action and potential as PZQ-alternatives for the control of schistosomiasis.

Some of the earliest work on MCLZ four decades ago demonstrated that the effect of this compound on parasites was very similar to PZQ [15]. Both compounds cause rapid parasite $Ca^{2+}$ influx, contractile paralysis, and tegument damage [15,16]. More recently, both PZQ and MCLZ have been shown to be agonists at related schistosome Transient Receptor Potential

(TRP) channels [17–20], providing a likely molecular explanation for this phenotypic similarity.

Despite both compounds acting as agonists at related $Ca^{2+}$ channels, there are differences in their antiparasitic effects. PZQ has broad-spectrum activity against many schistosome species, while MCLZ is active against African but not Asian parasite species [4,15]. TRP channels from these two clades of parasites display variation in the proposed benzodiazepine binding pocket, providing a straightforward explanation for this difference in drug response [19]. Radioligand binding experiments also indicate that the two compounds do not share binding sites on the parasite tegument [21,22], which would be consistent with the two compounds acting on different TRP channels. Finally, PZQ and MCLZ also display different effects on juvenile, liver-stage parasites. A notable drawback of PZQ is that it is unable to clear this stage of infection [1], which may contribute to lower than expected cure-rates in areas of high transmission [23]. On the other hand, MCLZ is effective against both liver-stage and adult parasites [4].

In this study, we have performed a detailed characterization of the *in vitro* and *in vivo* effects of MCLZ treatment on *S. mansoni*. There have been many studies describing the molecular effects of PZQ treatment on schistosomes [24–26]. But much less is known about the parasite response to MCLZ, since this lead has received comparatively little attention in the four decades since the two compounds were discovered. Detailed studies into the anti-schistosomal mechanism of benzodiazepines will provide a useful reference to compare similarities and differences with the current front-line therapy, PZQ.

## Materials and methods

### Ethics statement

Animal work was carried out adhering to the humane standards for the health and welfare of animals used for biomedical purposes defined by the Animal Welfare Act and the Health Research Extension Act. Experiments were approved by the Medical College of Wisconsin Institutional Animal Care and Use Committee (IACUC) protocols AUA00006471 and AUA00006735, IACUC of Texas Biomedical Research Institute (number 1419-MA), and UW-Madison School of Veterinary Medicine IACUC (protocol V006353).

### *In vitro* schistosome assays

The NIH-NIAID Schistosomiasis Resource Center (SRC) provided female Swiss Webster mice infected with *S. mansoni* (NMRI population, 180 cercariae per mouse). Animals were euthanized by $CO_2$ asphyxiation and cervical dislocation at either four weeks post infection to obtain juvenile, liver-stage worms or seven weeks post-infection to obtain adult, intestinal stage worms. Parasites were dissected from the host and cultured at 37°C/ 5% $CO_2$ in high-glucose DMEM supplemented with 5% fetal calf serum, penicillin-streptomycin (100 units/mL), HEPES (25 mM) and sodium pyruvate (1 mM). Worms were treated with either meclonazepam (Anant Pharmaceuticals) or DMSO vehicle control (0.1% vol/vol). Images of drug treated worms were acquired using a stereomicroscope and body length measurements were performed using ImageJ. Samples to be analyzed for pro-apoptotic caspase activation were stored at -80°C. Studies on PZQ-susceptible and PZQ-resistant schistosome populations (SmLE-PZQ-ES and SmLE-PZQ-ER) were performed in the same manner, except that Golden Syrian hamsters were used as hosts.

### *In vivo* schistosome assays

Schistosome infected mice provided by the SRC were used at either four weeks or seven weeks post-infection, depending on the developmental stage being studied. *S. mansoni* infected mice

were treated with MCLZ (30 mg/kg) or DMSO vehicle control at either four weeks or seven weeks post-infection (n = 5 mice per cohort). Mice were sacrificed by $CO_2$ euthanasia and cervical dislocation 14 hours later and worms were harvested from either the liver or mesenteric vasculature. Worms were immediately homogenized in TRIzol Reagent (Invitrogen) and stored at -80°C until ready for RNA extraction. Assays measuring the shift of adult parasites from the mesenteric vasculature to the liver were performed on mice harboring seven week old infections. Animals were dosed with MCLZ (30 mg/kg), and then euthanized as described above at various time points (n = 3 mice per time point) for counting the portion of worms in the liver versus the mesenteric vasculature. Samples to be analyzed for pro-apoptotic caspase activation were stored at -80°C until assays were performed, as described below.

### Caspase activity assays

Pro-apoptotic caspase 3/7 activation was measured using the Caspase-Glo 3/7 Assay Kit (Promega). Worms (pools of 5 male and 5 female worms) were homogenized in 125 μL of assay buffer (PBS + 0.3% triton X-100, supp

lemented with HEPES 10mM and Roche cOmplete Mini EDTA-free Protease Inhibitor Cocktail) and stored at -80°C. To perform the assay, worm homogenate was thawed, diluted 1:5 in distilled water, and then added to Caspase-Glo 3/7 substrate (1:1 volume ratio) in solid white 96-well plates. After 30 minutes, luminescence was read using a SpectraMax i3x Multi-Mode Microplate Reader.

### Electron microscopy

Worms were treated with MCLZ as described for *in vitro* schistosome assays, and then fixed overnight at 4°C in 2.5% glutaraldehyde/ 2% paraformaldehyde in 0.1M sodium cacodylate (pH 7.3). Worms were washed in 0.1M sodium cacodylate (3 x 10 minutes), post-fixed on ice in reduced 1% osmium tetroxide (2 hours), followed by washes in distilled water (2 × 10 minutes) and overnight staining with alcoholic uranyl acetate at 4°C. Samples were then washed in distilled water, dehydrated in MeOH (50%, 75% and 95%), followed by 10 minute washes in 100% MeOH and acetonitrile. Worms were incubated in a 1:1 mixture of acetonitrile and epoxy resin for 1 hour, prior to 2 × 1-hour incubations in epoxy resin. Samples were then embedded in epoxy resin (60°C overnight), and cut into ultra-thin sections (70 nm) onto 200-mesh copper grids and stained in aqueous lead citrate (1 minute). Imaging was performed on an Hitachi H-600 electron microscope with a Hamamatsu C4742-95 digital camera.

### Transcriptome sequencing

Total RNA was extracted from *S. mansoni* homogenized in TRIzol Reagent (Invitrogen) and libraries were generated using the TruSeq Stranded mRNA kit (Illumina) prior to sequencing using the Illumina HiSeq 2500 system (high-output mode, 50 bp paired-end reads at a depth of 20 million reads per sample). The RNA-seq pipeline was implemented using Nextflow [27] and is publicly available (https://github.com/zamanianlab/Core_RNAseq-nf). Trimmed reads (fastp, [28]) were mapped to the *Schistosoma mansoni* genome (v10, [29]) from Wormbase Parasite (release 19, [30]) using STAR [31]. Differentially expressed gene products between vehicle control and MCLZ-treated samples were identified using DESeq2 [32]. GO term analysis of differentially expressed genes was performed using g:Profiler [33] through Wormbase Parasite to identify enriched terms and pathways. Previously published RNA-Seq data on praziquantel exposed worms were retrieved for re-mapping to the updated *S. mansoni* genome (accession numbers PRJNA597909 and PRJNA602528, [26]). Transcriptomic data are contained in S1 and S2 Files. FASTQ files containing RNA-Seq data for control and

MCLZ treated worms are deposited in the NCBI SRA database (bioproject accession numbers PRJNA1150591 and PRJNA1177852). The MCLZ treated four week schistosome cohort (biosample accessions SAMN43291307, SAMN43291308, SAMN43291309, SAMN43291310, SAMN43291311) pairs with the four week cohort in bioproject accession number PRJNA597909 (SRR10776773, SRR10776772, SRR10776761, SRR10776760, SRR10776759). The MCLZ treated seven week data deposited under biosample accessions SAMN43291312, SAMN43291313, SAMN43291314, SAMN43291315, and SAMN43291316 pairs with the control seven week cohort deposited under biosample accessions SAMN43291317, SAMN43291318, SAMN43291319, SAMN43291320 and SAMN43291321.

## Bioinformatic analysis of divergent genes in parasitic flatworms

To identify *S. mansoni* genes with little homology to sequences in related, free-living flatworms, the *S. mansoni* predicted proteome (v10) was searched against a database of transcriptomes (tBLASTn) and predicted proteomes (BLASTp) from various related Platyhelminth species. Transcriptomes were retrieved from PlanMine [34] for *Prostheceraeus vittatus, Geocentrophora applanata, Gnosonesimida sp. IV CEL-2015, Protomonotresidae sp. n. CEL-2015, Prorhynchus alpinus, Rhychomesostoma rostratum, Stylochus ellipticus, Bothrioplana semperi, Dugesia japonica, Planaria torva, Polycelis nigra, Polycelis tenuis,* and *Dendrocoelum lacteum*. Predicted proteomes were retrieved from WormBase Parasite [30] for *Schmidtea mediterranea* (S2F19H1) and *Macrostomum lignano*. Divergent genes were defined as those *S. mansoni* gene products with no hits or low homology hits (e-value > 0.01) against other flatworms. The list of these divergent genes is provided in S3 File.

## TRP expression in single cell RNASeq data

TRP subunit expression was analyzed using a published adult *S. mansoni* single cell RNASeq dataset [35]. Sequences from TRP subunits identified in [36] were retrieved from WormBase Parasite (WBPS18, [30]) and aligned with MAFFT(v.7.471) [37] and trimmed using trimAI(v.1.4.1) [38] (gapthreshold=0.7, resoverlap=0.7, seqoverlap=0.7). The trimmed and filtered alignment was used as input for IQ-TREE 2(v.2.2.2.6) [39] to build a maximum likelihood tree with ultrafast bootstrap approximation (1000 bootstrap replicates) [40] for nodal support. Single cell co-expression of TRP channel subunits was calculated using the rcorr() function from the Hmisc R package [41].

# Results

## Meclonazepam exposure results in caspase activation and parasite death

The benzodiazepine MCLZ causes rapid, $Ca^{2+}$-dependant paralysis and tegument damage in *S. mansoni* that is very similar to the effect of PZQ exposure. However, the two drugs are thought to target distinct TRP channels with different cell and tissue expression. Therefore, we were interested in further investigating the molecular response of parasites to *in vitro* and *in vivo* MCLZ treatment to provide a comparison to what is known about the parasite response to the existing anthelmintic, PZQ. As expected, both juvenile liver-stage and adult worms treated with MCLZ (5 μM) display contractile paralysis (**Fig 1A**). Sectioning and imaging of these worms by transmission electron microscopy revealed damage to the parasite tegument, with extensive vacuolization of the tissue layer above the body wall muscle, as well as internal tissues within the parenchyma. The whole organism phenotype was apparent at concentrations of 3 μM and above (**Fig 1B**). Following overnight incubation in MCLZ, visual observation of parasite contraction corresponded to molecular indicators of cell death (measurement of pro-apoptotic caspases, Caspase-Glo 3/7 Assay System, Promega). But while MCLZ's

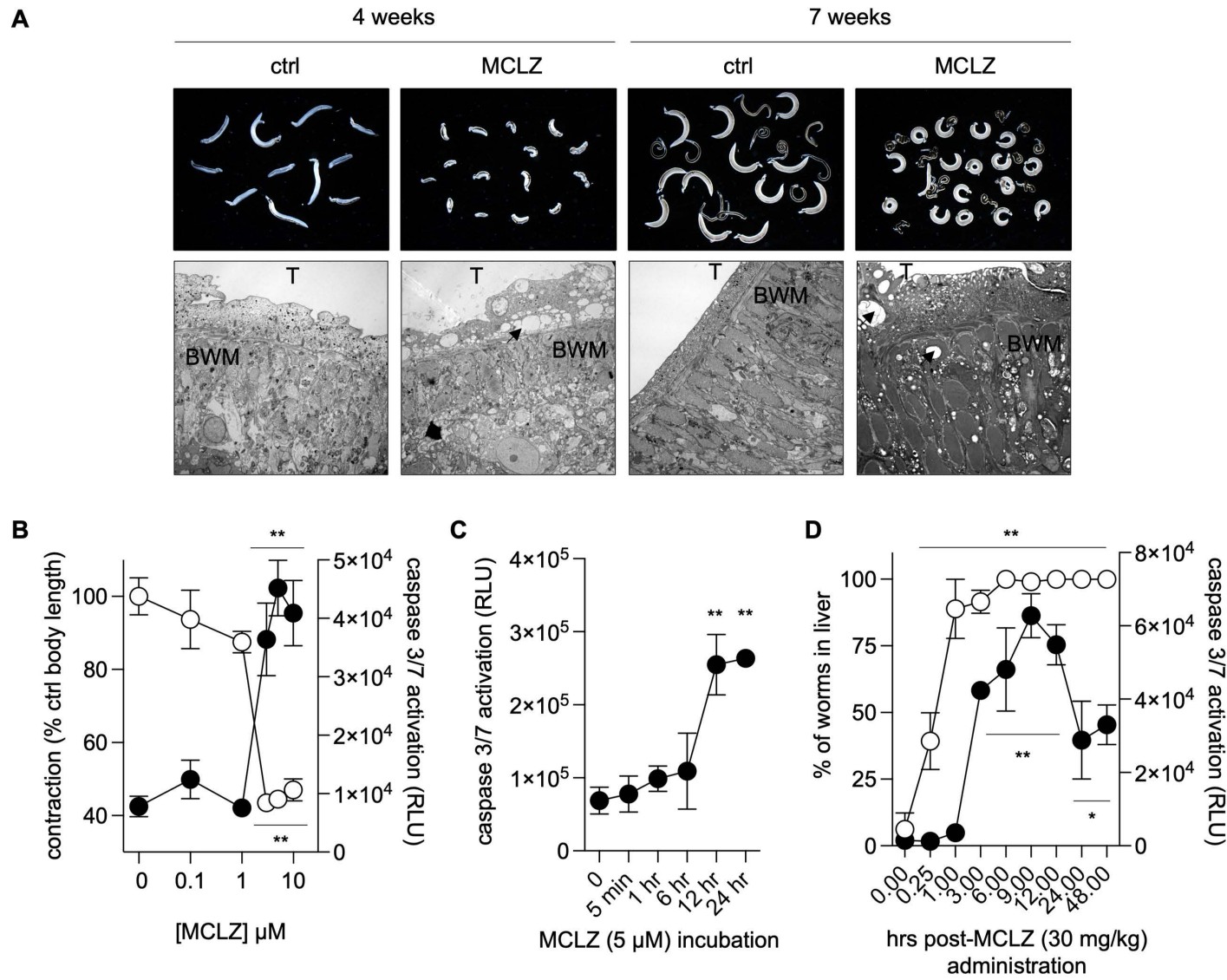

**Fig 1. Parasite tissue damage and death following meclonazepam exposure.** (A) Juvenile, liver-stage parasites (four week infections, left) and adult, intestinal stage parasites (seven week infections, right) exposed to meclonazepam (MCLZ 5 μM, 14 hours). Top - Brightfield images of worms fixed after drug treatment. Bottom - Worms were then sectioned and processed for imaging by transmission electron microscopy (TEM), showing damage to parasite tissues. Note vacuoles (arrowed) forming in the tegument layer (T), as well as sub-tegumental tissues at and below the body-wall muscle (BWM), of MCLZ treated worms. (B) *In vitro* concentration (0 - 10 μM) response for MCLZ treated adult parasites measuring contraction (open symbols, expressed as percent of DMSO control treatment) and activation of pro-apoptotic caspases 3/7 (solid symbols). Symbols denote mean ± standard error of **n** = 13 adult male worms for length measurements and **n** = 6 replicates of caspase activity assays for each treatment, each replicate consisting of ≥ 5 worms per drug concentration that were pooled into a single tube and homogenized. (C) Time course of pro-apoptotic caspase activation in adult parasites following *in vitro* MCLZ treatment (n = 6 replicates as in (B), ** p < 0.01 significance of difference relative to t = 0 hr treatment time point). (D) Time course of MCLZ effects on parasites following *in vivo* administration of 30 mg/kg dose of drug to mice harboring seven week infections. Mice (n = 3 per time point) were euthanized at various intervals (ranging from 15 minutes to 48 hours) to measure shift of worms from the mesenteric vasculature to the liver (open symbols) and caspase activation (solid symbols). All timepoints were assessed as significantly different from the **t** = 0 hr control ANOVA and Dunnett's multiple comparisons test, * **p** < 0.05 and ** **p** < 0.01.

contractile phenotype resulting in shortened body length is almost immediate, pro-apoptotic caspase activation does not occur until approximately 12 hours of *in vitro* drug exposure (MCLZ 5 μM, **Fig 1C**).

The *in vivo* effects of MCLZ treatment are similarly rapid. In the absence of drug, adult *S. mansoni* reside in the mesenteric vasculature. Very few or no worms are present in the

liver at this stage (seven weeks post infection). Following treatment with an anthelmintic, worms are paralyzed and swept from the mesenteric vasculature, through the portal vein, and into the liver. The timing of this can vary for different anthelmintics. For example, PZQ has been reported to cause this 'hepatic shift' within 6–24 hours, while another commonly studied antischistosomal drug, oxamniquine, takes up to six days to exert its effects [42,43]. To characterize the effects of MCLZ, mice infected with adult *S. mansoni* parasites were orally administered 30 mg/kg of drug, which in prior studies has been shown to be a curative dose [4,44]. Mice were then euthanized and dissected at various time points, collecting the worms from either the mesenteric vasculature to the liver. Within 15 minutes after drug treatment, parasites were already paralyzed and beginning to be swept to the liver (39.3 ± 10.7% of worms). This is the earliest time point we are able to reliably study given the time required for delivering drug to the mouse by oral gavage, $CO_2$ euthanasia, and then dissection of the liver and mesenteric vasculature, indicating that the impact of MCLZ is almost immediate. The parasites collected from each mouse were snap frozen for storage and then processed to assay pro-apoptotic caspase activation using the same Caspase-Glo 3/7 Assay as mentioned above. High levels of caspase activity were detected at three hours and beyond (**Fig 1D**). While this is slightly earlier than we have previously reported for PZQ [43], a direct comparison is difficult given these are two different compounds with different doses and potency. Nevertheless, in both cases activation of pro-apoptotic caspases occurs after the shift of worms from the mesenteric vasculature to the liver.

## Parasite transcriptional response to *in vivo* meclonazepam treatment

Numerous transcriptional studies have been performed on the response of *S. mansoni* to *in vitro* and *in vivo* PZQ exposure [24–26], but there are no reports on exposure of schistosomes to MCLZ. A unique feature of MCLZ is that it can cure infections at both the four week, juvenile liver-stage as well as seven week, adult parasites [4]. The juvenile stage is refractory to treatment with PZQ, as well as a range of other compounds that have been experimentally studied [45]. Therefore, we dosed mice harboring parasites at both the four and seven week timepoints with MCLZ (30 mg/kg) or DMSO vehicle control. Mice were then euthanized 14 hours later, and parasites were dissected from the livers and mesenteric vasculature, homogenized in TRIzol, and processed for RNA extraction and transcriptome sequencing. Read mapping data to version 10 of the *S. mansoni* genome (PRJEA36577) is provided in S1 File. Principal components analysis (S1 Fig) and distance matrix plots (S2 Fig) confirm clear separation of DMSO and MCLZ treated cohorts. Analysis of juvenile and adult stages shows 625 and 803 gene products, respectively, are differentially expressed between control and drug treated samples, when applying a filter of FDR adjusted p-value < 0.01 and $\log_2$ fold change > 1 (**Fig 2A**-**2C** and S2 File). There was relatively little overlap between these datasets, with only 25% of upregulated transcripts and 20% of downregulated transcripts shared between the juvenile and adult lists (**Fig 2C**). This degree of overlap is not unexpected, given the differences in the biology of the two life cycle stages (liver-stage, four week old worms are not sexually mature, while intestinal, seven week old worms are sexually dimorphic and have begun to mate and lay eggs).

What can the transcriptional response to MCLZ treatment tell us about this compound's mechanism of action? Gene ontology enrichment returns terms related to iron transport (e.g., ferroxidase activity, iron ion transport, intracellular iron ion homeostasis) for transcripts upregulated in juvenile worms. Terms such as $Ca^{2+}$ ion binding, peptidase inhibitor activity, microtubule-based process and dynein complex are enriched in the list of transcripts differentially expressed in adult worms, which is broadly consistent with what is known regarding MCLZ-evoked $Ca^{2+}$ influx and tegument damage. A closer look at the gene products with

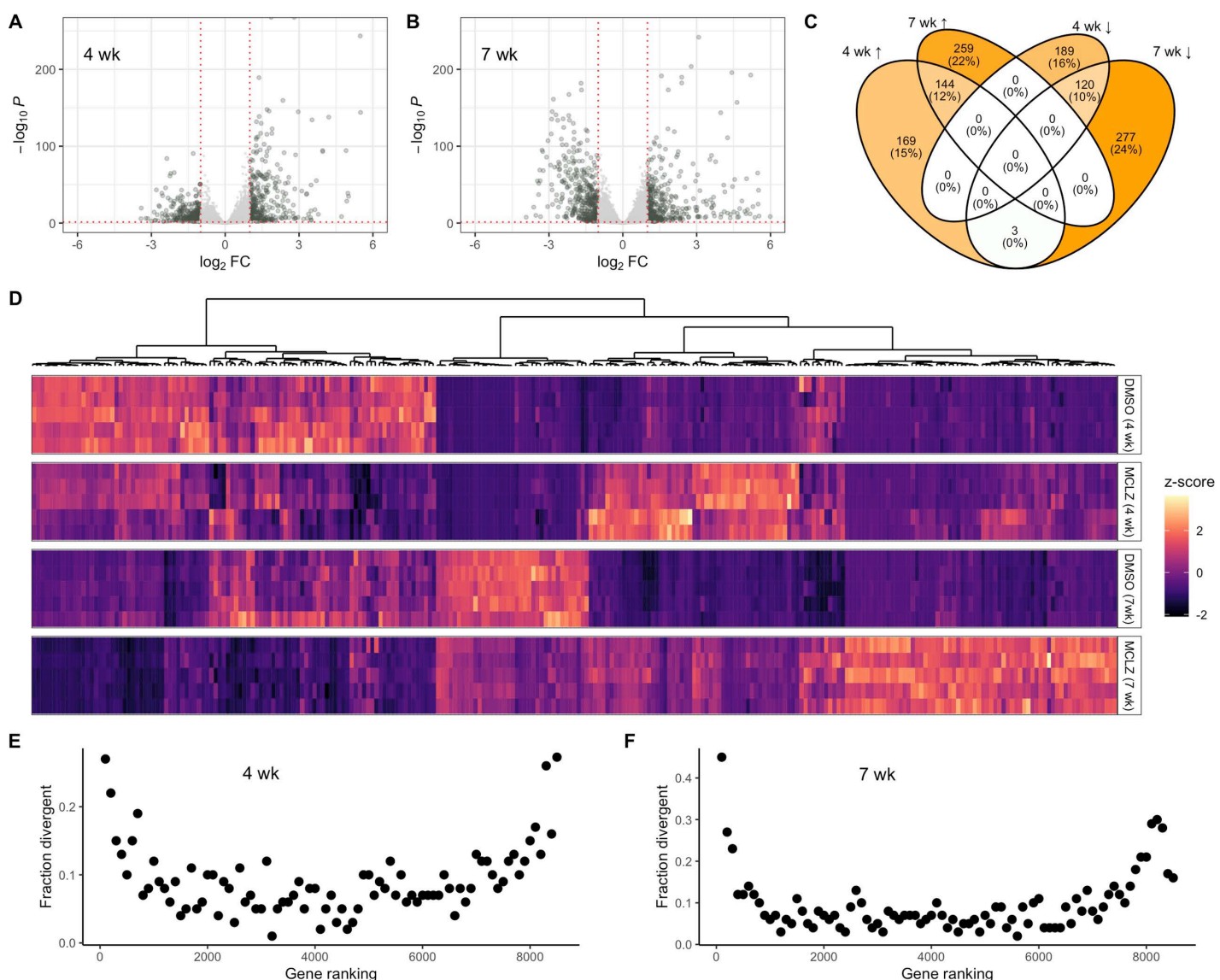

**Fig 2. Transcriptional response of parasites to *in vivo* meclonazepam treatment.** (A) Volcano plots of transcript expression in juvenile, four week parasites (left) and (B) adult, seven week parasites (right) harvested from mice treated with either DMSO control or MCLZ (30 mg/kg). Cut-offs for differentially expressed genes shown at $\log_2$ fold change > 1 and adjusted p-value < 0.01. (C) Venn diagram summarizing overlap between differentially expressed gene lists (↑ = upregulated, ↓ = downregulated) in four and seven week parasites. (D) Hierarchical clustering of differentially expressed, parasite divergent gene products across the four and seven week DMSO control and MCLZ treated cohorts. (E & F) Analysis showing that transcripts from genes that are more divergent in parasitic flatworms are also enriched in the extremes of up and downregulated rankings of **(E)** four week, juvenile and **(F)** seven week, adult parasites following MCLZ treatment. Gene ranking on the x-axis reflects ordering from positive (left) to negative (right) $\log_2$ fold change with MCLZ treatment. Each closed symbol on the scatter plot represents a window of 100 transcripts, and the y-axis reflects the proportion of these transcripts that are categorized as parasite divergent (BLAST e-value > 0.01 when querying free-living flatworm genomes or transcriptomes).

these annotations reveals several *S. mansoni* ferritin genes are upregulated with MCLZ treatment (Smp_311630 and Smp_311640 in both juveniles and adults; Smp_047650 in juveniles). Other potentially interesting differentially regulated transcripts include several TRP family members (TRPM ortholog Smp_000050 and TRPP ortholog Smp_334610). This is notable given that the putative target of MCLZ (Smp_333650, [19]) belongs to this family of ion channels, although this particular gene product was not differentially expressed. Gene products

with peptidase inhibitor activity annotation include five Kunitz-type protease inhibitors such as SmKI-1 [46]. Finally, enrichment of terms such as $Ca^{2+}$ binding and dynein complex is due to the presence of numerous schistosome tegumental-allergen-like (TAL) proteins in the list of downregulated adult transcripts (e.g., SmTAL 1, 2, 3, 5, 11, 12 and 13). These proteins contain two N-terminal EF-hand motifs and a C-terminal dynein light chain domain [47].

However, GO-term analysis may be of limited value in organisms such as schistosomes with poorly annotated genomes and many proteins of unknown function. A third of all differentially expressed transcripts (34% of juvenile and 32% of adult list) have no mapped GO terms. Similarly, many lack protein domain annotation (37% of juvenile and 35% of adult differentially expressed genes have no PFAM annotation). Analysis of up and downregulated transcripts shows enrichment for gene products that are highly divergent in parasitic flatworms relative to free-living flatworms (**Fig 2D**). When the predicted *S. mansoni* proteome is searched by BLAST against a database of 15 phylogenetically diverse free-living flatworms, 1482 of the 9920 genes in the WBP19 release return either no hit or a result with an e-value > 0.01 (S3 File), indicating these have little homology to gene products from free-living relatives. When looking at where these parasite divergent genes are distributed in a ranking of the differentially expressed genes, they are not distributed evenly across the dataset. 19% of juvenile and 26% of adult differentially expressed genes fall within this parasite divergent gene list, compared to just 8% of non-differentially expressed juvenile and adult genes. The effect is even more pronounced near the top of the differentially expressed gene lists (e.g., 45% of the top 100 upregulated adult transcripts are parasite divergent). This pattern is observed in both four week (**Fig 2E**) and seven week (**Fig 2F**) datasets. Therefore, we are faced not just with the challenge that many gene products in non-model organisms are of unknown function, but also that these are enriched in the lists of transcripts that change with drug treatment. Fortunately, some of these divergent sequences have been studied previously, allowing us to infer molecular changes occurring in response to MCLZ. For example, the most upregulated transcript in drug treated adults (Smp_138080) is a microexon gene (MEG). This gene family is unique to flatworms and expressed in tissues such as secretory cells and the tegument [48–50]. Many other members of the MEG family are in the list of juvenile and adult differentially expressed transcripts (20 out of the 39 MEG family members, S2 File).

## Comparing transcriptional changes over a time course of anthelmintic exposure

Given the phenotypic similarity of adult worms treated with MCLZ and PZQ, we wanted to compare the transcriptional response of the two drugs. Mice harboring seven week infections were treated with MCLZ (30 mg/kg) and sacrificed at various intervals (0 hour no drug time point, 15 minutes after drug, and 1, 3, 6, 12, 24 and 48 hours after drug; n=3 mice per time point). Worms were harvested from mice at each time point, pooled, and homogenized in TRIzol. Transcriptome sequencing and read mapping was performed, as with the four and seven week samples. Previously published time course transcriptomic data for *S. mansoni* harvested from mice treated with PZQ (400 mg/kg) [26] was also re-mapped to the newest version of the genome (read mapping data for both drugs provided in S1 File). For comparison, the doses of both of these drugs (MCLZ 30 mg/kg and PZQ 400 mg/kg) reflected the lowest completely curative oral dose in the murine model. Samples were then clustered based on gene expression profiles (**Fig 3A**), with earlier time points grouping together by drug treatment (e.g., 0.25 to 6-hour MCLZ treatments clustering together, distinct from PZQ treatments). The 12–24-hour time points then cluster separately, with PZQ and MCLZ drug treatments again grouping together, followed by the 48-hour timepoints. Differentially expressed

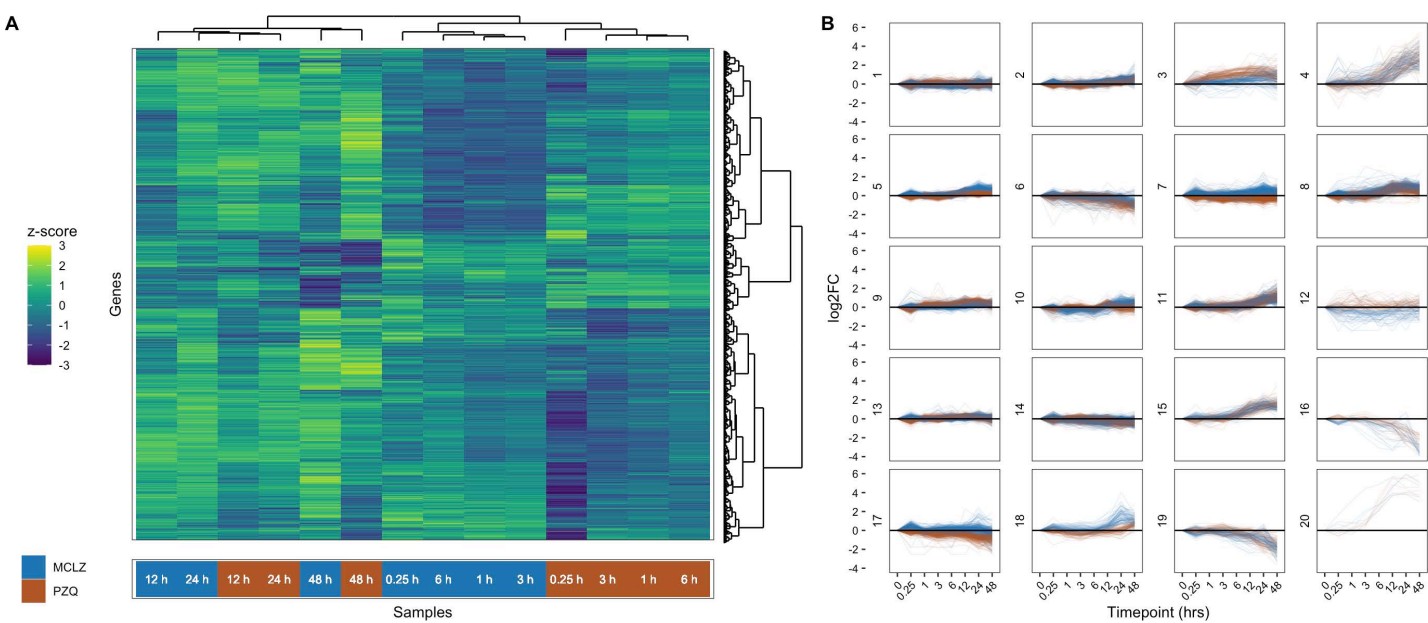

**Fig 3. Time course of gene expression changes following different anthelmintic treatments.** (A) Hierarchical clustering of seven week, adult *S. mansoni* harvested from mice treated with either a curative dose of MCLZ (30 mg/kg) or PZQ (400 mg/kg). Heatmap reflects z-score of gene expression (transcripts per million) across MCLZ and PZQ drug treatment cohorts. (B) Comparative trajectories of transcripts with similar and different short and long term expression changes following drug exposure. Blue = MCLZ treated samples. Brown = PZQ treated samples.

gene products identified at the 14 hour time point in **Fig 2B** were largely reproduced in this experiment (S3 Fig). The majority (291/403) of the upregulated differentially expressed genes from **Fig 2B** are represented in clusters 4, 8, 11 and 15 (**Fig 3B**), which have an increasing trajectory in the time course data. Similarly, the majority (255/400) of downregulated differentially expressed genes from **Fig 2B** are represented in clusters 6, 14, 16 and 19 (**Fig 3B**), which have a decreasing trajectory in the time course data. The transcriptional changes for MCLZ (blue line plots) were largely mirrored in the PZQ dataset (brown line plots) at later time points (12 to 48-hours).

## Schistocidal activity of meclonazepam on praziquantel-resistant worms

Given that MCLZ and PZQ exposure causes similar phenotypic changes and transcriptional responses in parasites, we were interested in whether the schistocidal effect of MCLZ would be altered in a laboratory population of PZQ-resistant *S. mansoni*. We used two populations, one that is PZQ-susceptible (SmLE-PZQ-ES, PZQ IC$_{50}$ ~600 nM) and one that is PZQ-resistant (SmLE-PZQ-ER, PZQ IC$_{50}$ > 200 μM), which have been selected from the Brazilian LE parasite population showing a polymorphic response to PZQ [14,20,51]. This difference in PZQ response is associated with differential gene expression in the TRP channel target of PZQ, Smp_246790. Adult male worms were treated with MCLZ at increasing concentrations (n = 30 worms each of MCLZ 0, 0.3, 0.5, 1, 3 and 5 μM) for 24 hours and then frozen, homogenized, and processed for testing using the Caspase-Glo 3/7 luminescent assay system. MCLZ-evoked caspase activation was comparable in both populations, reaching a maximum signal of 5.8 ± 0.6 and 6.3 ± 0.6 fold change over 0 μM control cohorts for the PZQ-susceptible and resistant populations, respectively (**Fig 4**). Estimates of the MCLZ EC$_{50}$ for each population are comparable (PZQ-susceptible population = 2.5 ± 0.7 μM, PZQ-resistant population = 1.4 ± 0.4 μM).

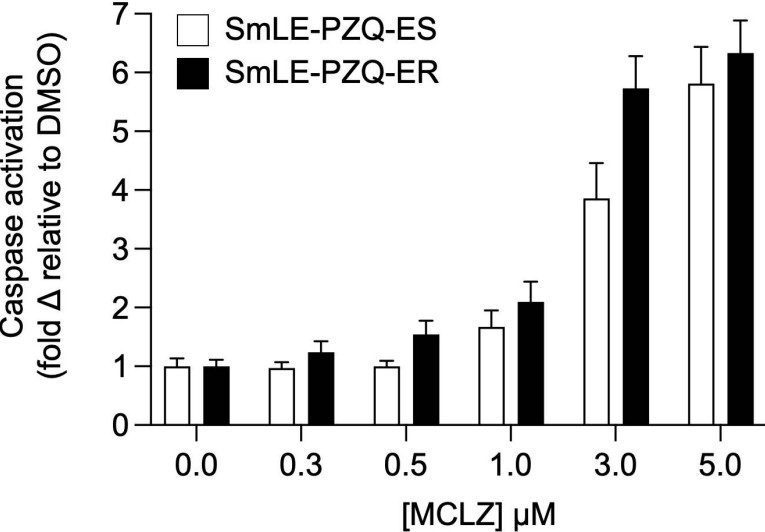

**Fig 4. Praziquantel-resistant parasites are susceptible to meclonazepam.** Pro-apoptotic caspase activation in seven week, adult *S. mansoni* treated *in vitro* with various concentrations of MCLZ (0 - 5 μM) for 24 hours (n = 30 worms per concentration for each schistosome population). Open bars reflect the average ± standard error response of the PZQ sensitive population SmLE-PZQ-ES, and solid bars indicate response of the PZQ-resistant population SmLE-PZQ-ER.

## Discussion

While the anthelmintic benzodiazepine MCLZ was discovered in the 1970's, there has been relatively little work done on this lead compared to the more extensive studies on the molecular effects of PZQ on schistosomes [24–26]. Historically, MCLZ was not pursued due to the dose-limiting sedative side effects typically associated with the benzodiazepine drug class [5,6,52,53]. However, sequenced genomes of parasitic flatworms show that schistosomes lack the $GABA_A$Rs that mediate host sedation [43,54,55]. Instead, benzodiazepines appear to act on a schistosome TRP channel [19]. This target may be the channel which mediates rapid, drug-evoked $Ca^{2+}$ influx [15]. Schistosomes possess a Bcl-2 regulated cell death pathway which activates pro-apoptotic caspases through mitochondrial outer-membrane permeabilization [56]. Mitochondria have an important function buffering cytosolic $Ca^{2+}$ levels, and sustained elevation in intracellular $Ca^{2+}$ promotes opening of the mitochondrial permeability transition pore, release of cytochrome c, and activation of pro-apoptotic caspases [57]. This class of compounds merits revisiting to assess its antischistosomal potential, given the difference in host and parasite benzodiazepine targets and findings that it is possible to modify meclonazepam to develop analogs with reduced sedative effects [44].

Several questions remain regarding the antiparasitic mechanism of benzodiazepines. To what extent does the mechanism of antischistosomal benzodiazepines mirror that of the existing frontline therapy, PZQ? Both compounds are proposed to act as agonists on parasite $Ca^{2+}$ permeable TRP channels (PZQ on Smp_246790, and benzodiazepines on Smp_333650). Is it the case that $Ca^{2+}$ influx through different channels drives distinct downstream events, resulting in different outcomes? Or do these initially distinct pathways converge on similar downstream signaling pathways and tissue types? Transcriptome sequencing of MCLZ treated worms enables a comparison with our previously published PZQ datasets to investigate some of these questions.

## Potential function of gene products differentially expressed following MCLZ treatment

Analysis of transcriptomic data shows that *in vivo* MCLZ exposure results in differential expression of different gene sets in juveniles and adults, although there is a subset which are shared between the two developmental stages (**Fig 2**). Many of these transcripts are divergent in parasitic flatworms, with poor homology to sequences in free-living flatworms. Bioinformatic prediction of the function of these individual gene products can be difficult, because gene annotations are commonly mapped from conserved genes studied in more conventional model organisms. But we can hypothesize that enrichment of divergent parasite gene products in the differentially expressed gene set reflects their function in regulation of the host-parasite interface. This would be consistent with MCLZ damage to the parasite surface, immune-recognition of parasites following chemotherapy, and the notion that gene products expressed at the host-parasite interface undergo more rapid evolution [58,59].

Several upregulated transcripts may reflect a schistosome evasion strategy following compromise of structures such as the tegument that typically cloak the parasite from the host immune system (**Fig 1A**). Two previously studied gene families include micro-exon genes (MEG) and Kunitz-type protease inhibitors. MEGs belong exclusively to parasitic flatworms and have been proposed as regulators of host-parasite interactions [48,50], with some genes perhaps generating variant antigens in a manner analogous to protozoan parasites [48]. Some members of the gene family have been explored as potential vaccine candidates [48,49,60]. These include the three members of the MEG-3 Grail family (MEG-3.1 or Smp_138080, MEG-3.2 or Smp_138070, and MEG-3.3 or Smp_138060), which are all in the top ten most upregulated transcripts in MCLZ-treated adults. Expression of these gene products has been proposed as a strategy for parasite immune-evasion [48]. Other members of the MEG family are downregulated with MCLZ-treatment. These include the MEG-4 gene product Sm10.3 (Smp_307220), which has been shown to be immunogenic and partially effective as a vaccine, and MEG-8 (Smp_171190), MEG-9 (Smp_125320) and MEG-12 (Smp_152630), which are seroreactive esophageal gene products in rodents, primate models and humans [61–64].

In other parasites and blood feeding invertebrates, Kunitz-type protease inhibitors have an anti-hemostatic and immunomodulatory function (e.g., ticks [65] and black flies [66]). There is evidence that the same is true in schistosomes. The most well studied member of this family is SmKI-1, which displayed anti-coagulant and anti-inflammatory properties *in vitro* [67] and is secreted by the parasite to inhibit host neutrophil recruitment *in vivo* [46]. In older *S. mansoni* genome releases SmKI-1 was mapped to Smp_147730, but this gene ID is deprecated and currently maps to three different genes, Smp_311660, Smp_311670 and Smp_337730. All three of these are upregulated in both juvenile and adult MCLZ-treated datasets. These same transcripts are also upregulated following *in vivo* PZQ exposure [26]. Up-regulation of these transcripts may be a protective parasite response to drug-induced tegument damage. This protein may also be recognized by the host immune system, as SmKI-1 has been shown to be partially effective as a vaccine candidate in murine models [68,69].

Just as Kunitz-type protease inhibitors may be of particular biological significance for blood feeding parasites, ferritins are also noteworthy in our MCLZ upregulated dataset. In other blood feeding invertebrates, ferritins have a role in managing oxidative stress resulting from ingested iron [70–72]. Schistosomes also encounter the same challenge of managing iron homeostasis [73], and dysregulation of these processes may account for the anti-schistosomal efficacy of many antimalarial drugs. Three schistosome ferritins have been experimentally studied, Scm-1 (Smp_087760) and Scm-2 (Smp_047650) which are expressed in the yolk and somatic cells, respectively [74], and a *S. japonicum* gene SjFer0, which is related to

Smp_063530 [75]. However, the two most dramatically upregulated ferritins in our MCLZ-treated datasets are unstudied ferritins, Smp_311630 and Smp_311640. Single cell transcriptomic data indicates these are expressed in tegument and flame cells. These gene products were also upregulated in response to PZQ in our prior study [26]. This may reflect a parasite response to oxidative stress [76], driven by disruption of ion homeostasis caused by either drug [15].

Parasites may also respond to drug-exposure by down-regulating expression of antigens that are recognized by the host immune system. This may be reflected in the transcriptional changes in genes encoding schistosome tegument-allergen-like proteins (SmTALs), which have been implicated in the host immune response to parasite infection. The predominant parasite antigen recognized by host IgE is SmTAL-1 (Sm22.6 or Smp_045200) [77,78], which is downregulated in MCLZ-exposed adult parasites. In total, eight different SmTALs are differentially expressed with drug exposure at the adult stage. This response seems to be specific to mature worms, as only one SmTAL is downregulated in drug-treated juvenile liver-stage worms.

These transcriptional responses implicate numerous gene products that have likely interactions with the host immune system. While several prior studies on PZQ demonstrated the importance of the host immune system in clearance of schistosome parasites following chemotherapy with this drug [42,79,80], similar experiments have not yet been performed with benzodiazepines. MCLZ appears capable of killing directly *in vitro* (**Fig 1B** **and 1C**), but it is not known whether the plasma concentration of the drug is sustained at levels necessary to eliminate parasites without contribution from the host immune system *in vivo*.

## Comparative response of adult schistosomes to MCLZ and PZQ treatment

These data raise several questions for further exploration regarding the mechanisms of action for MCLZ and PZQ. The two compounds differ most significantly in their efficacy against juvenile parasites, which may be explained by pharmacokinetic differences. PZQ has a half-life of 0.5 hours (mice) to 2 hours (humans) [81,82], while MCLZ has a half-life in humans of ~40 hours [83]. If, theoretically, PZQ were administered in a manner that extended it's half-life by 20 to 80 fold, it may also be effective at clearing juvenile worms.

The remarkable similarity between the phenotypes of PZQ and MCLZ on schistosomes - contractile paralysis, tegument damage, and similar transcriptional responses (**Fig 5**) - indicates similar mechanisms of action. Heterologous expression of the two TRPs activated by MCLZ and PZQ indicates that each compound is selective for their proposed target (e.g. PZQ, but not MCLZ, activates Smp_246790 [18] and benzodiazepines, but not PZQ, activate Smp_333650 [19]) when the receptors are expressed as homomeric channels. Data showing that PZQ resistant schistosomes with decreased expression of TRP Smp_246790 are still susceptible to MCLZ also provides *in vivo* support that the two compounds bind distinct targets (**Fig 4**). But tetrameric TRP channels may form functional heteromers, and one parsimonious explanation for the similar effects of the two compounds could be that they activate a common Smp_333650/ Smp_246790 heteromer. However, analysis of published adult single cell RNA-Seq data [35] does not indicate co-expression of these two subunits in the cells, although some of the cells expressing each TRP subunit do cluster together and may engage similar downstream pathways (**Fig 5**). Short term differences in transcriptional changes caused by the two drugs may reflect distinct changes resulting from activation of their respective targets, while similar transcriptional changes at later time points may reflect a more generalizable parasite response to engagement by the host immune system following parasite tegument damage.

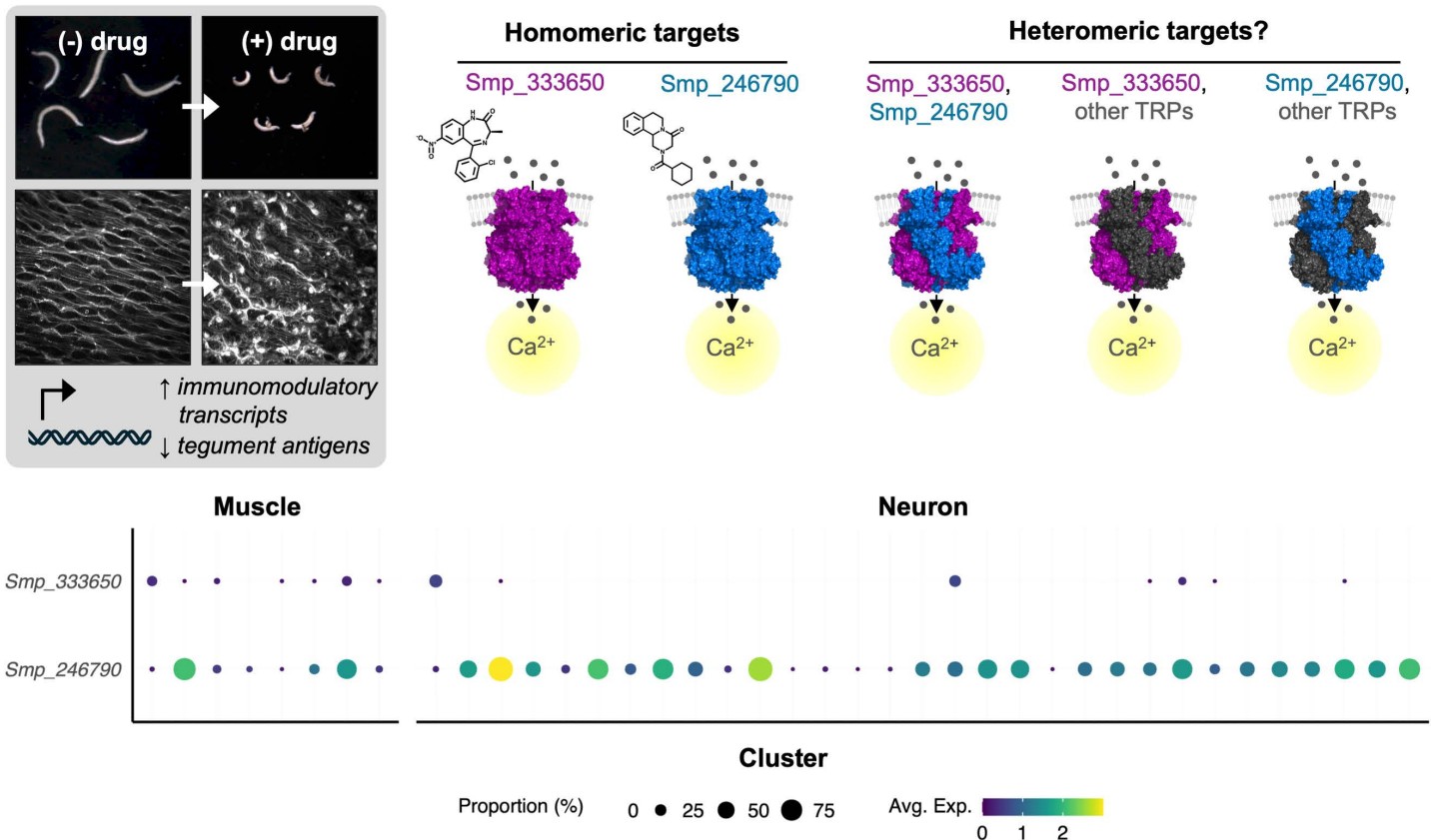

**Fig 5. Potential mechanisms of meclonazepam and praziquantel.** MCLZ and PZQ have nearly identical phenotypes on adult parasites (*left*), such as contractile paralysis and tegument damage, as well as similar transcriptional responses. Both compounds are agonists at heterologously expressed, homomeric TRP channels (*right*); MCLZ on Smp_333650 (purple) and PZQ on Smp_246790 (blue) [18,19]. Within the parasite, these TRP subunits could conceivably form either homomeric or heteromeric channels consisting of multiple subunits expressed in the same cell type. Analysis of single cell transcriptomic data [35] (*bottom*) for Smp_333650 and Smp_246790. Muscle clusters 1 - 8 are ordered left to right, as are neuron clusters 1 - 31. Dot diameter denotes the percentage of cells within the cell type cluster expressing the gene of interest. Dot color indicates the average expression of the indicated gene by cluster.

This study expands our understanding of *S. mansoni* transcriptional changes evoked by *in vivo* chemotherapy, providing insight in the comparative mechanisms of different drug treatments. Given the similar anthelmintic phenotypes between meclonazepam and praziquantel, and new meclonazepam analogs demonstrating decreased sedation [44], this chemical series is a promising route for research into new praziquantel-alternatives for schistosomiasis control.

## Supporting information

**S1 Fig. Principal component analysis of control and meclonazepam treated cohorts.** RNA-Seq data for four week (left) and seven week (right) infections treated with DMSO control (circles) or MCLZ (triangles).
(PNG)

**S2 Fig. Sample distance plot for four week and seven week control and MCLZ treated cohorts. Distance matrix for all samples.** Replicates cluster based on drug treatment condition and parasite developmental stage.
(PNG)

**S3 Fig. Time course of differentially expressed gene list following meclonazepam treatment. Gene expression = log$_2$ fold change in TPM relative to no drug, t = 0 hour time point.** Top = expression of the 403 upregulated gene list from **Fig 2B**. Bottom = expression of the 400 downregulated gene list from **Fig 2B**.
(PNG)

**S1 File. Read mapping data for drug treated schistosome samples. (Sheet 1)** Key with sample ID and metadata. **(Sheet 2)** Expression data in gene counts for all samples. **(Sheet 3)** Expression data in transcripts per million, TPM).
(XLSX)

**S2 File. Analysis of differentially expressed gene products between control and meclonazepam treatment.** DESeq2 output for DMSO and MCLZ treated samples. **(Sheet 1)** Gene expression data for juvenile, four week parasites. **(Sheet 2)** Gene expression data for adult, seven week parasites.
(XLSX)

**S3 File. List of parasite divergent gene products.** BLAST results searching the predicted proteome of the *S. mansoni* genome v10 against a database of free-living flatworms. **(Sheet 1)** tBLASTn results querying the *S. mansoni* proteome against a database of free-living flatworm transcriptomes (see methods for species information). **(Sheet 2)** BLASTp results querying the *S. mansoni* proteome against the *Schmidtea mediterranea* and *Macrostomum lignano* predicted proteomes. **(Sheet 3)** List of Smp IDs which either do not return a hit in either of these queries, or return hits with an e-value > 0.01.
(XLSX)

## Acknowledgments

Schistosome infected mice were provided by the NIH-NIAID Schistosomiasis Resource Center for distribution through BEI Resources, NIH-NIAID Contract HHSN272201700014I. Work at Texas Biomedical Research Institute was conducted in facilities constructed with support from Research Facilities Improvement Program grant C06 RR013556 from the National Center for Research Resources, and with the support of the Vivarium, which is part of the SNPRC at Texas Biomedical Research Institute supported by grant P51 OD011133 from the Office of Research Infrastructure Programs, NIH.

## Author contributions

**Conceptualization:** Winka Le Clec'h, Frédéric D. Chevalier, Mostafa Zamanian, John D Chan.

**Data curation:** Clair R Henthorn, Paul McCusker, Mostafa Zamanian.

**Formal analysis:** Clair R Henthorn, Paul McCusker, Mostafa Zamanian, John D Chan.

**Funding acquisition:** Timothy JC Anderson, Mostafa Zamanian, John D Chan.

**Investigation:** Clair R Henthorn, Paul McCusker, Winka Le Clec'h, Frédéric D. Chevalier, John D Chan.

**Methodology:** Clair R Henthorn, Paul McCusker, Winka Le Clec'h, Frédéric D. Chevalier, Mostafa Zamanian, John D Chan.

**Project administration:** Timothy JC Anderson, Mostafa Zamanian, John D Chan.

**Resources:** Timothy JC Anderson.

**Supervision:** Winka Le Clec'h, Frédéric D. Chevalier, Timothy JC Anderson, Mostafa Zamanian, John D Chan.

 

**Visualization:** Clair R Henthorn, Paul McCusker, Mostafa Zamanian, John D Chan.

**Writing – original draft:** John D Chan.

**Writing – review & editing:** Clair R Henthorn, Paul McCusker, Winka Le Clec'h, Frédéric D. Chevalier, Timothy JC Anderson, Mostafa Zamanian, John D Chan.

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
