## [Decision Letter · Decision Letter 0]

4 Feb 2025

PNTD-D-24-01843Transcriptional phenotype of the anti-parasitic benzodiazepine meclonazepam on the blood fluke Schistosoma mansoniPLOS Neglected Tropical Diseases Dear Dr. Chan, Thank you for submitting your manuscript to PLOS Neglected Tropical Diseases. After careful consideration, we feel that it has merit but does not fully meet PLOS Neglected Tropical Diseases's publication criteria as it currently stands. Therefore, we invite you to submit a revised version of the manuscript that addresses the points raised during the review process. Please submit your revised manuscript within 30 days Apr 05 2025 11:59PM. If you will need more time than this to complete your revisions, please reply to this message or contact the journal office at plosntds@plos.org.  Please include the following items when submitting your revised manuscript: * A rebuttal letter that responds to each point raised by the editor and reviewer(s). You should upload this letter as a separate file labeled 'Response to Reviewers '. This file does not need to include responses to any formatting updates and technical items listed in the 'Journal Requirements' section below. * A marked-up copy of your manuscript that highlights changes made to the original version. You should upload this as a separate file labeled 'Revised Manuscript with Track Changes '. * An unmarked version of your revised paper without tracked changes. You should upload this as a separate file labeled 'Manuscript '. If you would like to make changes to your financial disclosure, competing interests statement, or data availability statement, please make these updates within the submission form at the time of resubmission. Guidelines for resubmitting your figure files are available below the reviewer comments at the end of this letter. We look forward to receiving your revised manuscript. Kind regards, Bruce A. RosaAcademic EditorPLOS Neglected Tropical Diseases Jong-Yil ChaiSection EditorPLOS Neglected Tropical Diseases

Shaden Kamhawi

co-Editor-in-Chief

Paul Brindley

co-Editor-in-Chief

**Additional Editor Comments:**  The reviewers are overall positive about the manuscript. Please carefully consider the suggested revisions, including the suggestions to add a comparison to the Park et al previous publication, to clarify some methods and to add some discussion about possible mechanisms.**Journal Requirements:** 

4) Please ensure that the funders and grant numbers match between the Financial Disclosure field and the Funding Information tab in your submission form. Note that the funders must be provided in the same order in both places as well.

**Reviewers' comments:**  Reviewer's Responses to Questions

**Key Review Criteria Required for Acceptance?**

**Methods**

-Are the objectives of the study clearly articulated with a clear testable hypothesis stated?

-Is the study design appropriate to address the stated objectives?

-Is the population clearly described and appropriate for the hypothesis being tested?

-Is the sample size sufficient to ensure adequate power to address the hypothesis being tested?

-Were correct statistical analysis used to support conclusions?

-Are there concerns about ethical or regulatory requirements being met?

Reviewer #1: The methods are appropriate.

Validation of the change (qRT-PCR) in expression of some of the most important differentially regulated genes would be useful

Reviewer #2: A few minor corrections/comments for the methods section:

In vitro schistosome assays

- What was the concentration of meclonazepam and how many worms?

- What is the drug half-life; how was 14hrs chosen as endpoint?

Transcriptome sequencing

- “Trimmed (fastq) reads…”. Missing citation?

- Missing citation for the pre-print of Schistosoma mansoni genome AND for WormBase ParaSite which should be used per WormBase ParaSite's data use policy

Bioinformatic analysis of divergent genes in parasitic flatworms

- Playhelminth should not be italicised

**Results**

-Does the analysis presented match the analysis plan?

-Are the results clearly and completely presented?

-Are the figures (Tables, Images) of sufficient quality for clarity?

Reviewer #1: The results are clearly presented. The figures are adequate.

Reviewer #2: “…are unstudied ferritin isoforms, Smp_....”. These look to be different genes entirely, not isoforms. It would be interesting to see the location of these genes in the genome and whether they are part of an operon, although perhaps out of scope for this project. I believed they are referred to as “isoforms” in at least two locations in the paper (results and discussion).

Regarding TRPMs, I’m curious whether the transcripts you found to be differentially expressed are the same as those from Park et al. (https://doi.org/10.1016/j.jbc.2023.105528). They describes the activation of TRPM channel Smp_333650. I’d appreciate the author’s comments on how their results either corroborate or are different from this other study.

**Conclusions**

-Are the conclusions supported by the data presented?

-Are the limitations of analysis clearly described?

-Do the authors discuss how these data can be helpful to advance our understanding of the topic under study?

-Is public health relevance addressed?

Reviewer #1: Due to the nature of the study, the results are largely descriptive. What is/are the mechanism(s) resulting in worm death? Apoptosis is induced, but how is it induced?

Reviewer #2: The discussion is concise and well-written. It is supported by the data presented.

**Editorial and Data Presentation Modifications?**

Reviewer #1: (No Response)

Reviewer #2: Please use either "single-cell" or "single cell" throughout

Fig. 1B and Fig. 1D are not referenced in the main text

**Summary and General Comments**

Reviewer #1: The manuscript by Henthorn et al. discusses a study to understand the schistosome-killing mechanism of action of meclonazepam, a drug with good activity against worms, but serious off-target activities. The authors used phenotypic assessment, induction of apoptosis, and RNAseq to monitor MCLZ effects on worms. RNAseq was used to characterize changes in worm gene expression treated with MCLZ both in vivo and ex vivo. The experiments are well designed. The paper is well written. Comparisons with results from the group’s study, also published in PNTD, using similar approaches to study the worm’s response to PZQ are useful.

Has it been determined that immune-recognition of parasites following chemotherapy is important in the MOA of meclonazepam as it is for PZQ?

Both PZQ and MCLZ cause Ca influx in worms, through action on different Trp receptors. Would similar results be found if worms were cultured and treated in Ca-free media?

Adding references to the sentence on page 11 ‘Numerous transcriptional studies …’ would be helpful.

‘If, theoretically, PZQ were administered in a manner that extended _it’s_ half-life by 20 to 80 fold, it may also be effective at clearing juvenile worms.’ This is experimentally feasible: treat infected mice b.i.d. at 20, 21, and 22 days after infection.

Reviewer #2: Henthorn et al. present a thorough study looking at the phenotypic and transcriptional responses of juvenile and adult-stage S. mansoni to the benzodiazepine, meclonazepam. They provide an intriguing history of the testing and subsequent discounting of MCLZ as a broad-use anthelmintic in favour of praziquantel. The in vitro test confirms previously observed phenotypic responses of worms from MCLZ. This was followed by an in vivo experiment and follow-up transcriptional analysis of worms from mice treated with MCLZ. It is significant, but not unsurprising, that the most up- or down-regulated transcripts were from “parasite divergent” genes with unknown putative function. The authors discuss potential mechanisms of action, discussing the potential involvement of TRP, ferritin, and Kunitz-type protease inhibitor genes. The effects of MCLZ on PZQ-resistant S. mansoni was looked at, confirming its use in potential PZQ-resistant populations.

PLOS authors have the option to publish the peer review history of their article (what does this mean? ). If published, this will include your full peer review and any attached files.

**Do you want your identity to be public for this peer review?** For information about this choice, including consent withdrawal, please see our Privacy Policy .

Reviewer #1: No

Reviewer #2: No

 **Figure resubmission:**  While revising your submission, please upload your figure files to the Preflight Analysis and Conversion Engine (PACE) digital diagnostic tool, https://pacev2.apexcovantage.com/. PACE helps ensure that figures meet PLOS requirements. To use PACE, you must first register as a user. Registration is free. Then, login and navigate to the UPLOAD tab, where you will find detailed instructions on how to use the tool. If you encounter any issues or have any questions when using PACE, please email PLOS at figures@plos.org. Please note that Supporting Information files do not need this step. If there are other versions of figure files still present in your submission file inventory at resubmission, please replace them with the PACE-processed versions.**Reproducibility:**  To enhance the reproducibility of your results, we recommend that authors of applicable studies deposit laboratory protocols in protocols.io, where a protocol can be assigned its own identifier (DOI) such that it can be cited independently in the future. Additionally, PLOS ONE offers an option to publish peer-reviewed clinical study protocols. Read more information on sharing protocols at https://plos.org/protocols?utm_medium=editorial-email&utm_source=authorletters&utm_campaign=protocols

---

## [Editor Report · Decision Letter 1]

9 Mar 2025

Dear Dr. Chan,

We are pleased to inform you that your manuscript 'Transcriptional phenotype of the anti-parasitic benzodiazepine meclonazepam on the blood fluke Schistosoma mansoni' has been provisionally accepted for publication in PLOS Neglected Tropical Diseases.

Best regards,

Bruce A. Rosa

Academic Editor

Jong-Yil Chai

Section Editor

Shaden Kamhawi

co-Editor-in-Chief

Paul Brindley

co-Editor-in-Chief

The authors have addressed the minor concerns from the reviewers, and the paper is now acceptable for publication.

---

## [Editor Report · Acceptance letter]

Dear Dr. Chan,

We are delighted to inform you that your manuscript, "Transcriptional phenotype of the anti-parasitic benzodiazepine meclonazepam on the blood fluke Schistosoma mansoni," has been formally accepted for publication in PLOS Neglected Tropical Diseases.

Best regards,

Shaden Kamhawi

co-Editor-in-Chief

Paul Brindley

co-Editor-in-Chief
